# SCARE: A Novel Switching and Collision Avoidance pRocEss for Connected Vehicles Using Virtualization and Edge Computing Paradigm

**DOI:** 10.3390/s21113638

**Published:** 2021-05-24

**Authors:** Mauro Tropea, Floriano De Rango, Nicolas Nevigato, Luigi Bitonti, Francesco Pupo

**Affiliations:** 1Department of Informatics, Modeling, Electronics and System Engineering (DIMES), University of Calabria, Via P.Bucci 39/c, 87036 Rende, CS, Italy; mtropea@dimes.unical.it (M.T.); pupo@dimes.unical.it (F.P.); 2NTT Data, c.da Cutura, Via Spagna 50, 87036 Rende, CS, Italy; nicolas.nevigato@nttdata.com (N.N.); luigi.bitonti@nttdata.com (L.B.)

**Keywords:** VANET, Mobile Edge Computing (MEC), Connected Vehicle (CV), virtualization, collision avoidance, switching mechanism

## Abstract

In this paper, some collision avoidance systems based on MEC in a VANET environment are proposed and investigated. Micro services at edge are considered to support service continuity in vehicle communication and advertising. This considered system makes use of cloud and edge computing, allowing to switch communication from edge to cloud server and vice versa when possible, trying to guarantee the required constraints and balancing the communication among the servers. Simulation results were used to evaluate the performance of three considered mechanisms: the first one considering only edge with load balancing, the second one using edge/cloud switching and the third one using edge with load balancing and collision avoidance advertising.

## 1. Introduction

Mobile Edge Computing (MEC) [1] is a distributed and decentralized Information Technology (IT) architecture able to elaborate and store critical data locally and transmit all received/elaborated data to a data center on the cloud. In this way, the elaboration, storage and networking information is closer to the devices that generate data. The typical use case is that of Internet of Things (IoT) [2] devices and implementations and the Connected Vehicles (CVs) field [3], which often faces latency problems and lack of bandwidth and reliability, not addressable through the conventional cloud model. Here, the MEC architecture is able to reduce the amount of data to be sent to the cloud, processing critical data, sensitive to latency, at the origin point, via smart devices, or sending it to an intermediate server, located nearby; the less ’time-sensitive’ data can instead be transmitted to the cloud infrastructure or data center, to allow more complex processing, such as big data analysis. In this context, particular attention has to be addressed to congestion reduction (e.g., [4,5]), routing (e.g., [6]) and mobility prediction (e.g., [7]).

In this paper, we consider two important aspects of a vehicular network [8]: one regards handover issues for guaranteeing service continuity to vehicles inside the considered area taking into account also the Road Side Unit (RSU) load balancing and the other aims to provide a mechanism able to avoid collision among vehicles. Both considered aspects are based on the use of the MEC paradigm [9] in order to consider a critical point in VANET environment: the latency. In fact, in a vehicular scenario, the number of events that happens in a second can be very high, and then the possibility of communicate with devices near the road can contribute to avoid vehicle crashes. In particular, in this work, we propose a solution that considers RSUs implementing the MEC paradigm, able to communicate with each other and vehicles in order to allow a ready knowledge of any dangerous event, guaranteeing service continuity and collision avoidance.

RSUs are designed as elements able to perform resource management in the network and thus guarantee correct functionality to vehicles. Two different service continuity mechanisms were implemented in Java language in the RSU: the first one is based on the threshold Received Signal Strength (RSS) technique [10] and the second one tries to exploit the cell breathing mechanisms typical of UMTS networks [11]. Moreover, together with these mechanisms, a collision avoidance system is proposed that can utilize RSUs and the communication between vehicles and RSUs based on new connectivity technology (i.e., 5G) in order to make safer the drivers on the roads through assisted driving systems. A study on the human reaction time is shown in order to analyze how different human reaction times can affect the human behavior on the road bringing to a different impact on the number of collisions.

Furthermore, it is possible to use virtualization [12] with Docker containers, for developing lightweight and portable servers [13], and the container orchestrator Kubernetes [14]. Moreover, to manage efficiently the server containers, they are grouped in a Cluster orchestrated by Kubernetes software platform [15]. For the management of vehicular traffic, constrained requirements in terms of latency, safety and fault tolerance are required. For this reason, MEC is preferred to cloud architecture [9]. Furthermore, we choose to use containers instead of virtual machines, due to their lightness and easy management [16].

This paper is an extension of two previous works [17,18], concerning the study of the possibility of switching between cloud and edge servers in the road context and how this can affect the collision impact on a VANET environment. In particular, it unifies these papers by providing a more detailed description of the proposed ideas and showing how human reaction times are related with vehicles’ collisions. Moreover, a unified mechanism called SCARE (a novel Switching and Collision Avoidance pRocEss) is proposed to improve load balancing between cloud and edge servers and propagating in real-time collision avoidance warnings at the EDGE.

The paper is organized as follows. A literature overview on CV and edge computing and collision avoidance mechanisms in VANET is presented in Section 2. The reference scenario is presented in Section 3. Section 4 describes the design of the critical parameters: reaction and latency times. The proposed services continuity mechanisms are described in Section 5. A description of the collision avoidance mechanism is provided in Section 6. Some details on the software implementation are described in Section 7. Finally, the performance evaluation and conclusion are presented, respectively, in Section 8 and Section 9.

## 2. Related Work

In this section, we provide a quick overview on the state of art for edge computing and the collision avoidance mechanisms related to VANET is summarized in this section. Many research papers deal with the new paradigm called Internet of Vehicles (IoVs); for example, Ji et al. [19] presented a survey on IoV, highlighting architectural and application aspects. Moreover, blockchain technology has been attracting great interest in ITS environment due to its capability of achieving a decentralized system [20].

### 2.1. Connected Vehicles and Edge in VANET

The term Connected Vehicle (CV) is related to a new concept which sees all users involved (not only cars but also pedestrians, bikes, traffic lights, infrastructures, etc.) able to communicate with each other in order to exchange information for increasing road safety. It constitutes one of the new frontiers of mobile technology. It poses important challenges in terms of number of users managed, latency and bandwidth. The new technologies related to 5G and the possibility of transferring part of the intelligence present in the network to its borders (MEC) will be a key enabling component. Recently, many technological innovations have been proposed to realize VANET technology by using Vehicle-to-Cloud (V2C), Vehicle-to-Infrastructure (V2I) and Vehicle-to-Vehicle (V2V) communications. In the literature, different works exist that deal with these new topics (Connected Vehicle (CV), Mobile Edge Computing (MEC), etc.). Researchers have proposed many solutions for the challenges that these new technologies arise.

In [21], the authors presented a framework for delivering data/service to the CVs. Using the optimization framework, they studied costs and parameters for using optimal bandwidth and vehicles flow in a data/service delivery system.

In [22], the authors studied the traffic volume in vehicular environment exploiting edge computing assisted 5G-VANET. Their proposal concerns a hierarchical architecture for distributing data using a scheduling scheme able to efficiently schedule vehicular content distribution. Their simulation results demonstrate the goodness of the proposed scheme.

In [23], the authors studied the performance of a system considering different protocols such as MQTT, HTTP and CoAP in a vehicular context showing how the performance is at the edge or cloud network infrastructure. In particular, they provided empirical insights on the advantages that an edge–cloud-based service provisioning can provide in comparison to a centralized cloud-based approach.

In [24], the authors presented a survey on the designs of autonomous driving edge computing systems, with particular attention to the security issues in autonomous driving and on how edge computing can resolve these issues.

In [25], an exploration of edges for connected and autonomous driving is shown. The authors suggested performing data elaboration closer to vehicles, thus guaranteeing a reduced end-to-end latency. They showed the goodness of using a mobile/multi-access edge computing (MEC) over 5G telecommunications.

In [26], the authors proposed a context-aware edge-based packet forwarding scheme for vehicular networks employing on a fuzzy logic-based edge node selection protocol to find the best edge nodes in a decentralized manner. They used a reinforcement learning algorithm to optimize the last two-hop communications in order to improve the adaptiveness of the communication routes. They evaluated their proposal through extensive simulation experiments.

### 2.2. Collision Avoidance and Edge in VANET

Many papers exist in the literature on topics that regard the automotive environment based on the distribution and dissemination of data between vehicles in order to solve many different situations in the road context. This data sharing is fundamental for all those emergency situations in which the propagation of vehicles information can play a key role for solving different issues and avoiding possible car collisions [27,28]. Many of these works exploit the edge paradigm to guarantee better performance in vehicular environment thanks to the low-latency and high-bandwidth characteristics of this approach [29,30].

In [31], the authors reviewed a series of models and algorithms proposed for avoiding collision events in the roads. In particular, they focused on two types of common accidents: rear-end crashes and intersection collisions. They highlighted the main technologies used in this field: cloud computing, neural networks, 4G and 5G.

In [32] the authors dealt with the autonomous vehicle issues that raise the problem of collisions and how to avoid them. They proposed a decentralized framework named RACE (Reinforced Cooperative Autonomous Vehicle Collision AvoidancE) that exploits a multi-agent deep reinforcement learning model to be used during autonomous driving. They implemented the RACE framework in the widely used TORCS simulator and conducted various experiments to show the goodness of their proposal by comparing their approach with existing autonomous driving systems.

In [33,34], the authors proposed a MEC approach based on a cooperative Collision Avoidance service, called MECAV, which tries to preview road hazards through the use of vehicles sensors equipment. Their collision avoidance service is allocated in the MEC infrastructure, which receives and processes information sent by vehicles and selectively alerts other vehicles of a possible emergency event on the road. They implemented a demo of the MECAV system, composed of two vehicles, two WiFi access points and the CAV service running on two MEC servers to validate the proposal architecture and functionalities.

Other studies about the use of MEC and collision avoidance system can be found in [35,36], where the authors proposed a MEC-based collision avoidance system able to protect not only vehicles but also vulnerable users, such as pedestrian, from possible collision events. Their experimental results provide indications on the reliability of algorithms for two types of road events: car-to-car and car-to-pedestrian collisions. They provided analyses for scenarios where a human driver is considered and automated vehicles (characterized by faster reaction times) populate the streets.

Another important study about collision was presented by Abou Elassad et al. [37], who proposed a fusion framework based on machine learning models able to provide a collision avoidance system capable of performing a prediction in real-time traffic crash. In particular, the framework takes into account a dataset including vehicle kinematics, driver behavior and weather conditions and combines several classifiers in order to make the final prediction. In [38], the authors proposed a new risk assessment-based decision-making algorithm able to guarantee collision avoidance in multi-scenarios for autonomous vehicles. In particular, the proposal taking into account driving style preferences (e.g., aggressive or conservative) and uses Carla simulator for evaluating and validate their approach.

### 2.3. Main Paper Contributions

In this work, we focus our attention on the a vehicular scenario, where two mechanisms are proposed for allowing vehicles switching among RSUs in order to guarantee RSUs load balancing and for making secure the considered system trying to avoid as many collisions as possible in the considered area. In particular, this work is an extension of two previous ones [17,18] published in conferences.

The first paper [17] deals with the problem of guaranteeing vehicles service continuity throughout the use of MEC technology. It provides two mechanisms of vehicles handover in the covered area showing how a better distribution of vehicles under the coverage of each RSU can provide better performance of the system at the expense of system latency time.

The second paper [18] presents a proposal of a collision avoidance mechanism that, exploiting edge technology, is able to switch between cloud server when the required constraints allow this type of communication characterized by high latency time. It shows how the use of 5G communication technology guarantees better performance of the system thanks to the low latency guaranteed by this technology.

In this paper, starting from these two works, we provide a merged study that analyzed the system behavior considering both the switching mechanism for vehicles handover process and the collision avoidance mechanism for the safety of drivers in a VANET environment balancing the communication among the cloud and edge servers. In particular, the main contributions of this work are summarized as follows:We propose two service continuity mechanisms able to guarantee in the handover phase the continuity of communication of all vehicles inside the considered VANET scenario. We show that the use of a cell breathing-like approach is able to better guarantee a load balancing of RSU covered areas.We propose a collision avoidance mechanism able to guarantee a low number of collision inside the VANET scenario, providing a method for utilizing a cloud or an edge RSU server on the basis of the latency time computed as human reaction plus the communication latency time.We analyze the contribution of the human reaction time in the vehicles scenario showing how times of reaction subdivided in three different categories can affect the response of the driver in the road context.The goodness of the proposed approach is proved by a set of simulation results in terms of latency time, switching number, collision number, collision percentage and server utilization percentage.We compare an edge-based mechanism versus the SCARE proposal in terms of collisions percentage and server utilization rate, in order to show how an approach taking into account load balancing and real-time advertising propagation is the right compromise in a emergency events scenario.

## 3. Reference Scenario

In this work, the considered scenario is a multi-layered architecture composed of three different parts: the vehicle layer, where there are vehicles traveling on the road; the edge layer, where there are RSU devices able to cover a specific portion of the map; and the cloud layer, where there is a centralized server able to perform different complex tasks (Figure 1). The roadside RSUs communicate with each others through the backbone infrastructure. Each RSU is equipped with an on-board processor for elaborating data, and for making decisions on the vehicles management. By taking advantage of 5G networks, latency is reduced to a few milliseconds [39]. A brief characteristics comparison between MEC and Cloud computing is shown in Table 1 [23].

When a problem occurs, the RSU evaluates whether to propagate messages arriving from previous RSUs towards the next ones or to stop forwarding because it is not necessary. All vehicles registered with the RSU are tracked; if the last vehicle has to be notified of a problem, then the message must be propagated to the adjacent RSU; instead, if the last vehicle is too distant from the event, then the message is not forwarded.

In this work, without loss of generality, it is assumed that all vehicles travel in one direction. To facilitate the simulation and make it faster, the two servers are supposed to be at a distance of 100 m from each other and their coverage range is 90 m. In this way, it is possible to know what happens during switching mechanism since the vehicles remain in the area covered by both nodes for a sufficiently long time (see Figure 1). Figure 1 shows three different zones: *Zone 1* is covered by first RSU; *Zone 2* is where a mobile device can be covered by both RSUs, here the switching mechanism takes place; and *Zone 3* is covered by only second RSU.

### Messages in the Scenario

In the considered scenario, the main actors are RSUs and vehicles that are able to exchange information by the use of specific messages. Five different typologies of messages exchanged between vehicles and servers were defined. The messages format is depicted in Figure 2. It is possible to view the main fields that the message transports in order to allow the correct functionality of proposed switching mechanisms.

In Table 2, a description of different message typologies used for the device communication is provided.

The messages exchange can allow a switching process that causes the change of RSU covered area. In this work, we propose two different switching mechanisms: one based on RSS threshold and the other one on a cell breathing-like approach typical of UMTS networks. We describe the two mechanisms and compare them to make some considerations about them.

## 4. Critical Parameters Design

For *latency*, in our work, we refer to the connectivity technology used in the vehicular context. In particular, we considered a 5G technology used by vehicles for communicating with the RSU placed on the road side (MEC server) or with the cloud server. The choice of communication technology clearly determines a greater or smaller latency time to take into account in the considered experiments.

### 4.1. Reaction and Latency Time

In road terminology, the term reaction time usually indicates the space and time, simultaneously necessary, for the driver of a vehicle to perceive a situation to be faced (in most cases, it is a danger that arises suddenly), realize perception in the mind and react with direct voluntary actions, with sufficient awareness of them, to a specific purpose; these actions and reactions, which take place in the context of the reaction time, are of a sensory, psychic, muscular and mechanical nature.

The duration of thereaction time is variable in relation to the psycho-physical conditions of the driver, the desired maneuver, the road and environmental conditions, the degree of efficiency of the vehicle and its speed.

The duration of the reaction time can be considered to be between approximately 0.75 and 1.50 s; in particular cases, it can be even greater [40,41]. In fact, from a general point of view, the duration of time $t_{r}$ can be considered variable in relation to the following typical conditions:perception and reaction of *Fast Reaction (FR)*, $t_{r}=0.75$ s;perception and reaction of *Normal Reaction (NR)*, $t_{r}=1$ s; andperception and reaction of *Slow Reaction (SR)*, $t_{r}=1.50$ s.

From when the vehicle driver sees an obstacle, he takes 1 s before starting to brake and travels a reaction space $\Delta s_{r}=v$· (1 s) which depends on the speed *v* at which he proceeds. If the obstacle is less than $\Delta s_{r}$ meters from the car, the driver does not even have time to start braking and collides with the obstacle with speed *v*. In Table 3, different speeds and spaces of reaction are shown.

It is as if the vehicle is $\Delta s_{r}$ meters longer and its length increases with speed.

#### 4.1.1. Braking Distance

The braking distance is the distance a vehicle travels between the start of deceleration and stopping. In the fairly realistic hypothesis that the deceleration produced by the brakes is constant, the motion of the vehicle is uniformly accelerated. The initial speed $v_{0}$, the final speed v=0 m/s, the acceleration −a and the braking distance $\Delta s_{f}$ are linked by the relationship (Equation 1):(1)$$\Delta s_{f}=\frac{1}{2}\frac{v^{2}-v_{0}^{2}}{a}=\frac{1}{2}\frac{v_{0}^{2}}{a}$$

The braking distance depends on the condition of the vehicle and the road surface, which determine the value of the deceleration −a. However, the important thing is that the braking distance increases with the square of the speed. For a car in good condition, on a road with medium grip, the braking distances are very similar to the values indicated in Table 4 and shown in Figure 3.

#### 4.1.2. Safety Distance

The safety distance is the distance that a vehicle must maintain from the one in front of it in order to stop without hitting it. The safety distance $\Delta s_{s}$ is the sum of the reaction space and the braking distance:(2)$$\Delta s_{s}=\Delta s_{r}+\Delta s_{f}$$

For example, at a speed of 90 km/h, we have: $\Delta s_{s}=25m+52m=77m$.

#### 4.1.3. Latency Time

The latency time concerns the communication time with cloud or edge server. The vehicle is able to decide if it should communicate with one of the two servers on the basis of the traffic condition in order to guarantee avoiding collisions. It is expressed in milliseconds (ms) and depends on the distance between vehicle and server location. The cloud server is remotely placed, while the MEC server is located in the road side unit (RSU) on the road side and, thus, near the vehicle and able to guarantee a lower latency in the communication. In our study, the latency time was set to 1 ms, considering as communication technology 5G [42,43]. In particular, this value was set equal for the communication with both server typologies.

### 4.2. Collision Condition Computing

To know latency and reaction times to be used for the collisions avoidance phase, first the design of some system parameters that characterize the considered scenario need to be accounted.

In the considered system, the vehicle is identified by a number called identifier (ID). The first vehicles of the queue is indicated with ID=0, while the vehicle behind has identifier ID=1. The superscript in our formulation indicate the vehicle identifier. Moreover, before describing the mathematical formulation of the considered motion equations, whose symbols are defined in Table 5, we considered some assumptions:constant speed of vehicles on the road “*v*”;same deceleration value “*a*”; anddistance *d* among vehicles.

Considering that suddenly the first vehicles brakes abruptly and indicating with $t_{lr}$ the latency time as sum of latency and reaction times needed to the vehicle behind to start braking, the following motion equation for vehicle with ID=0 at $t_{0}=0$ is considered:(3)$$\left\{\begin{array}{cc} x_{0}^{0}-\frac{1}{2}at^{2}+v_{0}^{0}t & t<\frac{v}{a}\\ x_{0}^{0}+\frac{1}{2}\frac{{\left(v_{0}^{0}\right)}^{2}}{a} & t\geq \frac{v}{a} \end{array}\right.$$

Instead, for vehicle with ID=1 at $t_{0}=0$, the motion equation is as follows:(4)$$\left\{\begin{array}{cc} x_{0}^{1}+v_{0}^{1}t & t<t_{lr}\\ x_{0}^{1}+\left(v_{0}^{1}+at_{lr}\right)t-\frac{1}{2}a\left(t^{2}+t_{cr}^{2}\right) & t\geq t_{lr} \end{array}\right.$$

Under the assumption that, due to an unexpected event, the first vehicle ID=0 suddenly brakes and the vehicles path are the same, we can calculate the collision time between the two vehicles as follows:(5)$$x_{0}^{0}+\frac{1}{2}\frac{{\left(v_{0}^{0}\right)}^{2}}{a}=x_{0}^{1}+\left(v_{0}^{1}+at_{lr}\right)t-\frac{1}{2}a\left(t^{2}+t_{lr}^{2}\right)$$

By computing, we obtain the following second-degree equation:(6)$$\frac{1}{2}at^{2}-\left(v_{0}^{1}+at_{lr}\right)t+\frac{1}{2}at_{lr}^{2}+d+\frac{1}{2}\frac{{\left(v_{0}^{0}\right)}^{2}}{a}=0$$

This equation has solutions when the delta (Δ) is greater than 0, that is:(7)$$\begin{array}{cc} \Delta ={\left(v_{0}^{1}+at_{lr}\right)}^{2}+4\left(\frac{1}{2}a\right)\left(\frac{1}{2}at_{lr}^{2}+d+\frac{1}{2}\frac{{\left(v_{0}^{0}\right)}^{2}}{a}\right)>0\\ \; & ⟹t_{lr}>\frac{d}{v} \end{array}$$

If Equation (Equation 7) is satisfied, that is the $t_{lr}$ time is greater than the ratio between considered vehicles distance and speed, we can consider that a collision happens between the vehicles when the first one suddenly brakes.

## 5. Services Continuity Proposed Mechanisms

In this section, two proposed switching mechanisms for providing vehicles’ services continuity are described. The first one considers the RSS threshold value in order to determine the time of switching towards the new RSU device coverage. The second one, on the basis of a cell breathing-like approach typical of a UMTS network, is able to perform a switching mechanism with load balancing for avoiding overload of RSU devices.

### 5.1. RSS Threshold Mechanism

This mechanism is based on messages received by RSU from vehicles under its coverage and on RSS value in order to perform the switching between the RSU stations [44]. It is depicted in Algorithm 1.

At the beginning, the RSU must know some information about RSU neighbors in order to communicate with them. Figure 1 shows that a vehicle inside *Zone 2* is able to receive a signal from RSU1 while in *Zone 3* it receives only the signal from RSU2.

Shortly, the RSU behavior:Upon receipt of a *Type 0* message, the RSU receives this message from the vehicle and becomes aware of a new vehicle in its area coverage. Thus, if an alert event happens, it can warn all coverage vehicles near this event and other RSUs with a *Type 1* message, informing about emergency situation.Upon receipt of a *Type 1* message, the RSU receives this message from the RSU and is aware of emergency situation and must check the presence of vehicles under its control near the vehicle braking sharply in order to avoid possible accident situations.Upon receipt of a *Type 2* message, the RSU receives this message from the vehicle and learns about switching completion by vehicle passed under its coverage. As a consequence, the RSU sends a *Type 3* message, containing vehicle ID, to the adjacent RSU to notify this change. Thus, the RSU can store this new condition in its database and no messages are more exchanged with this vehicle.

Clearly, this mechanism does not take into account load balancing behavior.
**Algorithm 1:** RSS Control Mechanism.
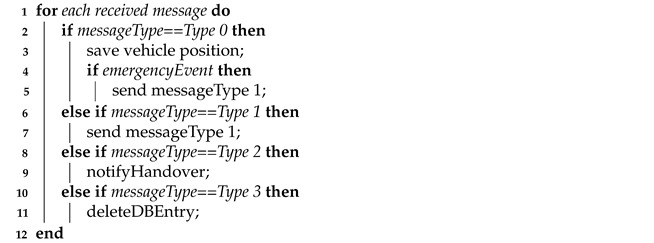


### 5.2. Cell Breathing-Like Mechanism

An enhanced mechanism was considered in RSUs to improve the switching mechanism modifying the RSU behavior in the coverage intersection zone (*Zone 2*). The enhancement is based on the cell breathing concept [11], typical of UMTS networks. The idea is to perform a sort of load balancing into the RSU devices, subdividing their load in terms of managed vehicles. Three new data structures were introduced:*VehiclesCommonZone*: It stores the number of vehicles in *Zone 2*, where potentially both RSU devices are able to manage vehicles.*BlackList*: It manages vehicles that, even if they are still in the coverage area of a RSU, this one does not manage these vehicles because assigned to other RSU.*VehiclesToControll*: It stores vehicles that have been assigned to the new RSU.

Upon receiving a message from a vehicle, the RSU verifies whether its ID is on its vehicles list:Vehicle not in the list: The RSU discards the message without further analysis since the vehicle is managed by another RSU.Vehicle in the list: The RSU analyzes the position of the vehicle and controls if it can potentially be transferred to another RSU by controlling position and coverage range.

Once the list has been updated, the RSU controls the maximum number of connections it can manage for vehicles belonging to *Zone 2*. Upon exceeding the limit, the RSU virtually decreases its coverage passing a vehicle control to the adjacent RSU, so to relieve its computational load, by sending a *Type 4* message. Thus, the new RSU can store vehicle ID in its database. This enhanced mechanism behavior, using the cell breathing-like paradigm, and its time diagram are reported in Algorithm 2 and Figure 4, respectively.
**Algorithm 2:** Cell breathing-like Control Mechanism.
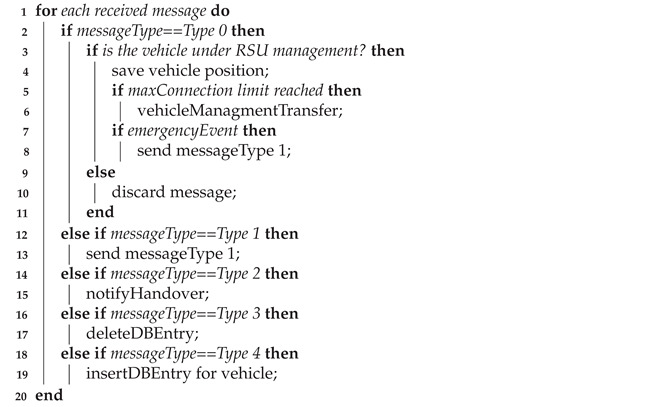


Figure 5 shows the different vehicles management with the two different proposed approaches.

## 6. Collision Avoidance Proposed Mechanism

In the following section, the proposed collision avoidance mechanism is described showing, through algorithm and flow chart representations, the behavior of servers and vehicles. Each vehicle is a CoAP client able to communicate with the server through the POST method exploiting Californium library [45]. The information about the vehicle status (i.e., ID, speed and position) is enclosed in a message. Furthermore, every command that vehicles receive from the server is managed by the simulator through a Java class, which implements vehicle behavior, as shown in Figure 6.

### Dynamic Collision Avoidance Mechanism

The main characteristic of this mechanism is to avoid collision among vehicles taking into account the load of the edge and cloud server to avoid of overloading the edge server by using the cloud when possible. In particular, the decision of which server to be used is made by the vehicle on the basis of traffic conditions with the aim of reducing the number of collisions. In particular, this decision is based on the calculated distance among a considered vehicle and its neighbors.

A vehicle, for the choice of the appropriate server, considers the total latency and reaction time in a scenario where vehicles have the same speed and deceleration values, as shown in Section 4.1. If the ratio of the distance between two vehicles over the speed is less than cloud communication latency plus vehicle reaction, then the vehicle will communicate with the edge to avoid collision; otherwise, it chooses to communicate with the cloud. This dynamic algorithm behavior is described in Algorithm 3.
**Algorithm 3:** Dynamic Switching behavior.
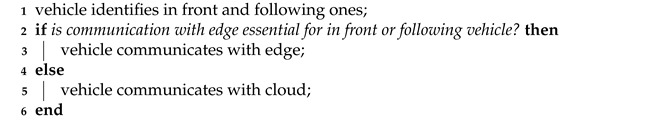


The main task of the MEC server implemented on the RSU is, upon receiving data from vehicles, to process this information in order to verify the presence of a danger or emergency braking by a vehicle, and then alert all following vehicles to avoid collisions. In particular, the RSU calculates the difference between two consecutive speeds of a vehicle, and, if it is greater than a certain threshold, a dangerous condition is identified and an alert is sent timely to all vehicles in a certain range to avoid collision. The operating principle is described briefly in Algorithm 4.
**Algorithm 4:** MEC server behavior.
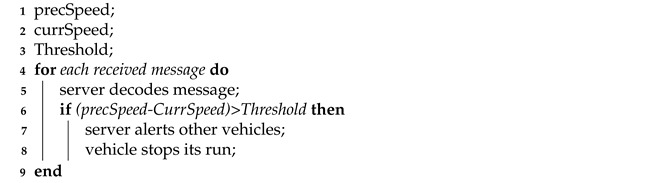


## 7. Software Implementation

To implement switching processes and collision avoidance mechanism, it was necessary to create two servers that work in parallel, able to simulate two different RSUs. This was applied considering three virtual machines able to form the Kubernetes cluster. One machine was used as a master, the other two as worker nodes. The virtual machines were hosted on a normal PC, as shown in Figure 7.

### 7.1. Virtualization Environment: Docker and Kubernetes

The simulated system was realized through the use of normal PCs that simulate the RSU units running on Docker container [46] managed by Kubernetes [47] and the vehicular traffic by using SUMO simulator [48] (see Figure 7).

Finally, the implementation of the MEC server and, therefore, the communication from server to vehicle was created. The final system can simulate vehicles in traffic that communicate with each other and the server to manage a collision avoidance mechanism, thus guaranteeing security in the vehicular environment. Vehicles periodically, by a device installed on board, send messages that contain information about their status towards the RSU MEC server. In particular, the message contains the identification number (ID) and the position and speed of the considered vehicle. Messages exchanged between vehicles and RSUs are processed and used to understand when an emergency braking is taking place, and therefore anticipate any collisions. In the case of emergency event, the MEC server generates an alert message and immediately sends it to the following vehicles within a certain range. When the alert message reaches the vehicles, the braking system is automatically activated to stop the vehicle safely and avoid a collision. A dynamic switching algorithm was also implemented that allows the vehicle to instantly decide whether to send the message to the MEC or cloud server, in order to avoid of overloading the devices on the roadside when it is not necessary.

Docker is an open-source project that automates the process of deploying applications within software containers, providing an additional abstraction thanks to virtualization at the Linux operating system level [46]. It is based on isolation features of Linux kernel known as *cgroups* and *namespaces* that make possible the creation of independent *containers* that can coexist on the same Linux instance, with the great advantage of being more lightweight and simpler then virtual machines. The main characteristic of Docker software is the absence of an hypervisor layer, typical of the virtual machine, and thus the possibility of being executed directly on the host machine.

Kubernetes is an open-source container orchestration and management system [47]. Initially developed by Google, it is now maintained by the Cloud Native Computing Foundation. It works with many container systems, including Docker. It is composed of two different components called masters and nodes, also called workers. They coordinate to run the workload on the servers that form a cluster controlled by Kubernetes that guarantees there is no downtime in a production environment.

The cluster is the core of Kubernetes, because it offers the ability to plan and run containers in a group of machines of any type: physical or virtual, on premise or in the cloud. It has at least one master and one worker node. The master is the component that provides tasks to worker nodes. The smallest and simplest object of a cluster is called a *pod*. A *pod* groups together containers that share storage and network resources and run on the same node. Another important object of Kubernetes is represented by *service*. It defines how to expose pods on an internal or external network. The service defines a name that is resolved by the cluster-internal DNS with one of the pods associated with it. The pods associated with the service are those that share the label defined by the service. By default a service is exposed inside a cluster, but it can also be exposed outside the cluster.

### 7.2. Vehicle Traffic Simulator

An urban mobility simulator, SUMO, was used to generate data from vehicles. SUMO is an open source software that allows creating various traffic scenarios [49]. Sumo can interact with the Java programs through *TraaS* library [50,51,52] making it easier to collect vehicle data at runtime. The simulator runs on a different PC, as shown in Figure 3. Eclipse was used as a development environment, able to interface with the simulator through the *SumoTraciConnection* library [48]. To carry out the communication between vehicles and MEC server and between vehicles and Cloud server, the CoAP protocol [53] was chosen using the Californium library written in Java [54]. Each vehicle was created as a CoAP client, and, periodically, makes requests to the MEC or Cloud server acting as CoAP server.

The vehicular traffic management is made by Sumo simulator where a stretch of road of *SS107 Silana-Crotonese*, in southern Italy, was considered and some XML files were created containing all information regarding the road portion such as the IDs of the two lanes of the roadway with the information of their direction, the maximum speed in m/s, etc. Moreover, the XML files contain the vehicles traveling information such as acceleration and deceleration values, ID, length, maximum speed in m/s and minimum gap with the previous vehicle, etc. Sumo allows importing the road network via *Open Street Map*, simplifying the process of developing the mobility model. This real world road network can be easily downloaded and modified and imported into the simulator through the *netconvert* command. In this way, a realistic vehicular traffic in the urban environment is generated. The Eclipse development environment is used to manage the simulator using code written in Java language and the library called *SumoTraciConnection* [48]. To start the simulation, a XML configuration file with the *.config* extension was created containing: the file related to the road route, the file related to vehicles and an additional file to display an alert on the map in case of a collision between two vehicles. Moreover, the configuration file contains a time section where the start and the end of the simulation with steps are specified. It is also possible to generate an output file with the simulation logs. The macro-diversity mechanism, typical of UMTS [55], is managed by simulator that allows vehicles to be connected simultaneously to different RSUs, so to guarantee vehicle connection also during the switching phase. Through the mechanism described above and the exchanged message, a vehicle is able to pass under the control of a new RSU.

### 7.3. Server Implementation

Servers were implemented in Java language and use CoAP protocol [53] to communicate with vehicles and each other. For the management of switching mechanism, two paradigms were used. The first one consists of the technique of RSS with threshold and the second one uses a combination of RSS with threshold and cell breathing. Moreover, to avoid collisions among vehicles, it processes the receiver data andnotifies timely all vehicles of the dangerous event.

## 8. Performance Evaluation

In this section, we present the results achieved using the simulator environment described in Section 3 and represented in Figure 1. It consists of two RSUs, covering a portion of road in which vehicles travel, and a cloud server for managing data information sent by vehicles. Different simulations were performed varying some input parameters and showing results in terms of number of packets, latency in milliseconds, switching number, collision percentages and server utilization. In Table 6, the main parameters used in the simulation experiments are summarized.

### 8.1. Switching Mechanism Analysis

In the following, we report graphics related to the proposed switching mechanisms showing the considered output parameters: packet number, latency time and switching number.

#### 8.1.1. Packet Number Analysis

A first result is the packets number managed by two RSUs that communicate with vehicles on the road, without cell breathing-like approach (Figure 8a) and with cell breathing-like mechanism (Figure 8b). In the first case, the management of vehicles and packets is entirely covered by the first RSU. This is because vehicles are at a point where the *RSS* parameters is above the minimum allowed threshold, and therefore no switching mechanism is performed. In the second case, instead, even though the vehicle positions are the same, a better load distribution can be appreciated. This is because some vehicles, although still in the range of the first RSU, are forced to move under the management of the second RSU.

In this simulation campaign we considered five different vehicle values (2–6 vehicles). The load distribution can be seen with the introduction of the fifth vehicle. This is due to the maximum number of established connections set to three to guarantee a load balancing among six vehicles. When there are still four vehicles in the system, the first vehicle of the set of vehicles is under the management of the second RSU, while the other three are still under the management of the first one. Therefore, the second RSU has no vehicles to notify, but merely informs the first RSU of any warnings. By adding another vehicle, a configuration with two vehicles managed by the second RSU is possible to observe, while three of five remain under the management of the first one. In this case, it is possible to view a small load distribution, because the first RSU has to manage three vehicles while the second one has two vehicles.

#### 8.1.2. Latency Analysis

In this paragraph, the latency analysis is provided. In particular, we show the elapsed time between the detection of the danger event and its notification to the last vehicle considering both switching techniques.

Figure 8c shows the system behavior considering the RSS approach. It is possible to observe a mean latency time of about 40 ms with a standard deviation of about 4.9 ms. The value of 40 ms represents the time needed to: send a message by vehicle to RSU, process it by RSU and send alert to all other vehicles. We observed that the number of involved vehicles does not contribute to increase latency time because the experiments show that vehicles receive the alert almost simultaneously, with a time lag of less than 1 ms. Figure 8d, instead, shows the behavior considering cell breathing approach. In this case, RSU1 and RSU2 control half of vehicles in the map respectively. In these experiments, we measured a mean latency time of about 60 ms with a standard deviation of about 4.6 ms. We observed that this mean value is increased because the message has to cross another hop (RSU) for reaching vehicles.

The results show a significant increase in latency if the cell breathing-like technique is adopted. This is due to the fact that, to notify the last vehicle, the message must go from the vehicle that generated the event to RSU 2, from RSU 2 to RSU 1 and, finally, from RSU 1 to the last vehicle.

#### 8.1.3. Switching Number Analysis

Finally, the last result for the switching analysis shows the number of switching per time unit considering into the map a number of five vehicles. The ratio between the switching number over the switching time has been calculated, obtaining the Switching Rate (SWR) per seconds (SWR/s). This value is affected by vehicles speed and by the distance from the vehicle ahead. Figure 8e shows the behavior of the switching mechanism. It is possible to observe that vehicles, increasing their speed, generate an increase number of switching given that they change more frequently their position. Moreover, it is possible to note that, at the same vehicle speed, the switching number per seconds increases when decreasing the distance between vehicles.

### 8.2. Collision Avoidance Analysis

In this section, we present the experiments’ results of the proposed collision avoidance system. Experiments were performed considering a scenario composed of a different number of vehicles in the map traveling with a speed of 50 km/h, the same deceleration values, a distance *d* from each other and reaction times in the range shown in Table 6 [56].

We provide experimental results of a collision analysis showing first a study about the number of collision considering different human reaction times and then a study on the collisions percentage in the considered scenario varying the distance between vehicles and between servers and emergency event point.

#### 8.2.1. Collision Number Analysis

To show the collision analysis considering human reaction times, in the following, the results about thenumber of collisions in the considered scenario varying the number of vehicles, the vehicles speed and traffic congestion are shown in Figure 9. The study aimed to indicate how the driving distraction, resulting in a slow human reaction, brings to a high number of collisions. In the next experiments, we considered different human reaction times and different numbers of vehicles with other parameters listed in Table 6.

Figure 9a gives some details about system behavior varying the vehicles number between 2 and 10. In particular, the analysis shows two important aspects: first how the collision number varies when in the map the number of vehicles increases and second how the human reaction time affects the increase of collisions. The so-called normal reaction time causes about 16% more collisions than the fast reaction time. Figure 9b gives other important indications about the number of collisions when the vehicle speed increases until arriving at typical highway speeds. Finally, Figure 9c shows indications of how congestion affects the number of collisions and how the human reaction times are related to it.

#### 8.2.2. Collision Percentage Analysis

The obtained results on collisions percentage are shown in Figure 10. Figure 10a shows details about the collisions percentage varying the vehicles distance. It is possible to note that with MEC server we have a lower percentage compared to the case with communication towards the cloud server. In addition, considering the same distance, MEC communication is able to guarantee better performance in terms of collisions percentage. Only by increasing the vehicles distance does the performance between server communications achieve similar values. Figure 10b shows the percentage of server (MEC and cloud) utilization evaluated considering three different conditions of congestion. The graphic shows how the aim of our proposal is respected; it allows, especially in certain vehicular traffic conditions, not to overload the MEC and communicate with the cloud server which has higher performance. In particular, when traffic is congested, the vehicles communicate more with the MEC server than with the cloud one since low latency times are required to avoid collisions. The figure shows that the MEC device is used a lot. If the traffic is not very congested, the utilization rate of the two servers is almost the same. On the other hand, considering non-congested traffic, each vehicle communicates with the cloud since the MEC server is not essential. Therefore, in this last case, only the cloud is used, as shown in the figure.

### 8.3. Edge vs. SCARE Approach Comparison

In Section 8.1, we emphasize the approach using load balancing at the edge, as it can reduce the server CPU usage percentage at the expense of a higher latency in the communication and advertising propagation.

Moreover, in Section 8.2, we show that, in the case of emergency situations, the communication with the edge server is more suitable than cloud because it permits alower collisions number.

Now, in this section, we present the SCARE approach that takes advantage of both previous mechanisms: the load balancing at edge of Section 8.1, where we show how a load balancing approach permits to avoid overloading the RSU resources, and the EDGE mechanism in the advertising propagation of Section 8.2, where we prove how the use of edge server in the management of traffic with high priority such as alert messages highly sensitive to the latency communication is more useful then cloud one. The SCARE mechanism joins the highlighted characteristics of the previous techniques in order to allow a reduced vehicles collisions number guaranteeing a better usage of edge servers’ resources.

Thus, Figure 11 shows how the edge with cell breathing-like mechanism and warning propagation, the SCARE approach, can perform in comparison with the approach of the edge with warning propagation but without RSU load balancing, the EDGE one. It is possible to see that, in the case of congestion, the SCARE approach can improve the performance of the server load, but it can increase the collisions percentage because some extra message delay due to the additional involved RSU. So, the warning condition is not respected producing collisions among vehicles. However, when the road congestion is medium-low, “little congested scenario” in the figure, the SCARE approach improves its performance for both collision and server usage percentage, guaranteeing lower values for both output parameters. In the last case, where no congested roads are considered, both approaches perform in a similar way because the RSU load balancing is not activated and the warning can be propagated in the single RSU respecting the threshold time for the car collision.

## 9. Conclusions

In this paper, we propose a vehicular architecture where the MEC paradigm is used and implemented in RSU devices exploiting the low latency MEC characteristic. The implemented system is based on virtualization concepts and uses Docker and Kubernetes in the development of the reference scenario.

The paper proposes a twofold approach. Firstly, it deals with the study of the services continuity issue proposing two different approaches based on the handover mechanism. A mechanism based only on RSS value is compared with an approach that takes into account a cell breathing-like mechanism in order to show how it is possible to balance the RSUs load in the common zone, avoiding of overloading RSUs. The drawback is a latency that increases with the number of vehicles introducing delay in the messages delivery. Then, a collision avoidance mechanism based on cloud and edge paradigms is shown. The paper describes how to use and communicate with a cloud or edge server in order to exploit the low latency MEC characteristic, fundamental in a scenario where communication times are strict requirements. The proposal shows that vehicles can use the cloud server in normal condition and the edge one in emergency situation in order to avoid overloading the edge server normally equipped with small resource in comparison to the cloud. Finally, on the basis of these two approaches, we present a joint approach, called SCARE, that takes advantage of both mechanisms and is able to allow a reduction in the vehicles’ collisions number and a better usage of the edge servers’ resource. The performed analysis gives some indications about system behavior in terms of collision number and percentage. Further, the paper gives analyses of how human reaction times can affect the collision behavior, providing useful indications on three different human reaction responses and how the driving distraction results in a slow human reaction that brings to a high collisions number.

## Figures and Tables

**Figure 1 sensors-21-03638-f001:**
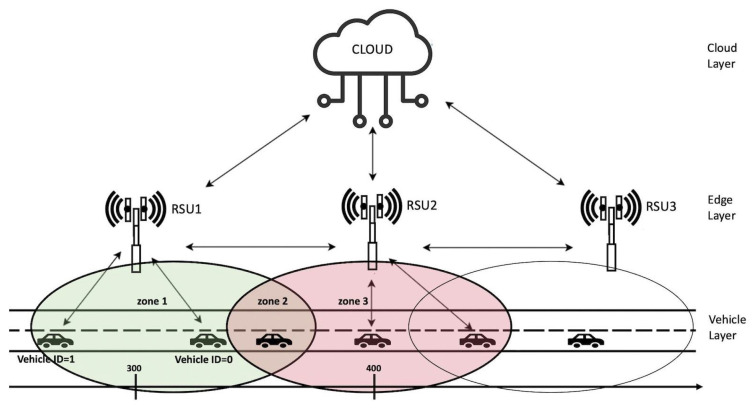
Reference Scenario.

**Figure 2 sensors-21-03638-f002:**
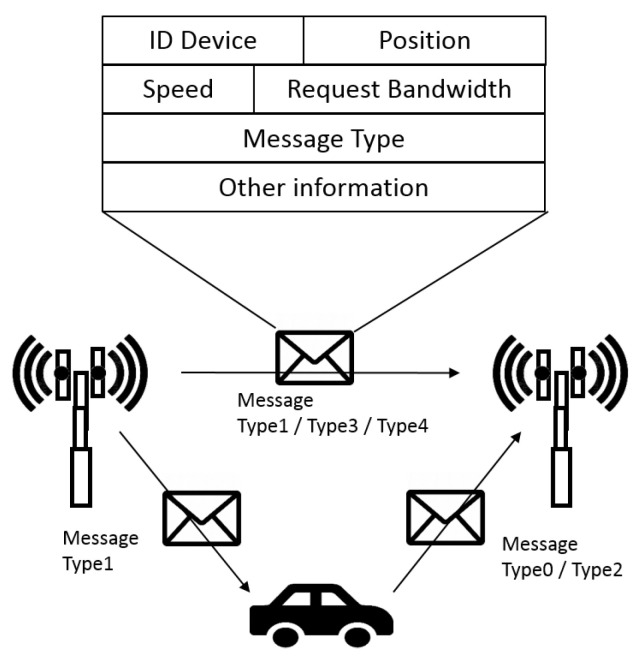
Message format exchanged among devices.

**Figure 3 sensors-21-03638-f003:**
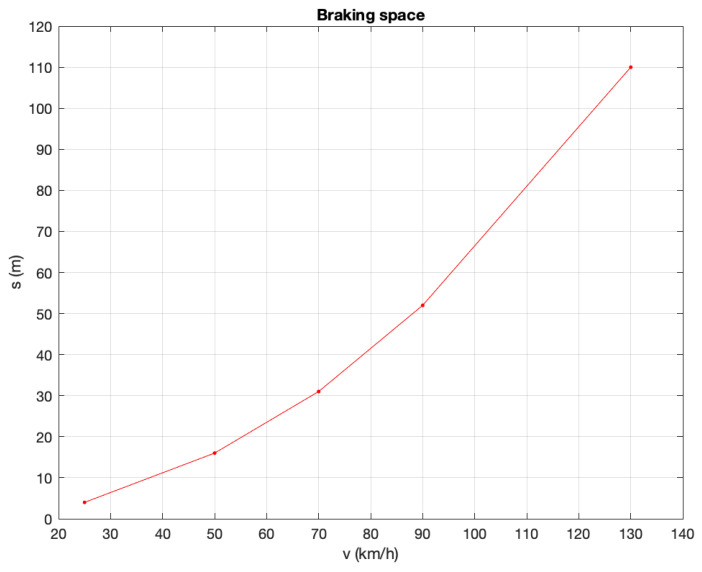
Braking distance (m).

**Figure 4 sensors-21-03638-f004:**
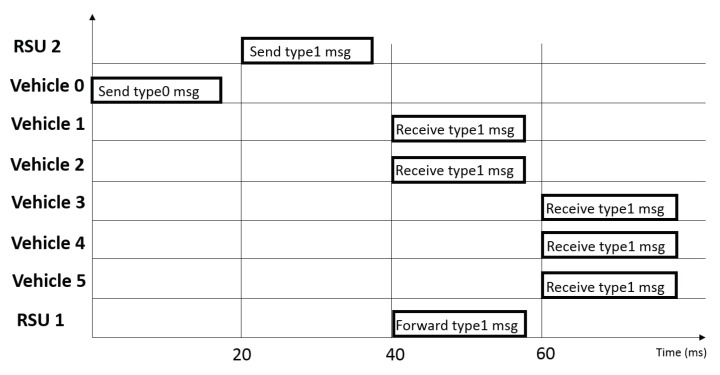
Time diagram of cell breathing-like mechanism.

**Figure 5 sensors-21-03638-f005:**
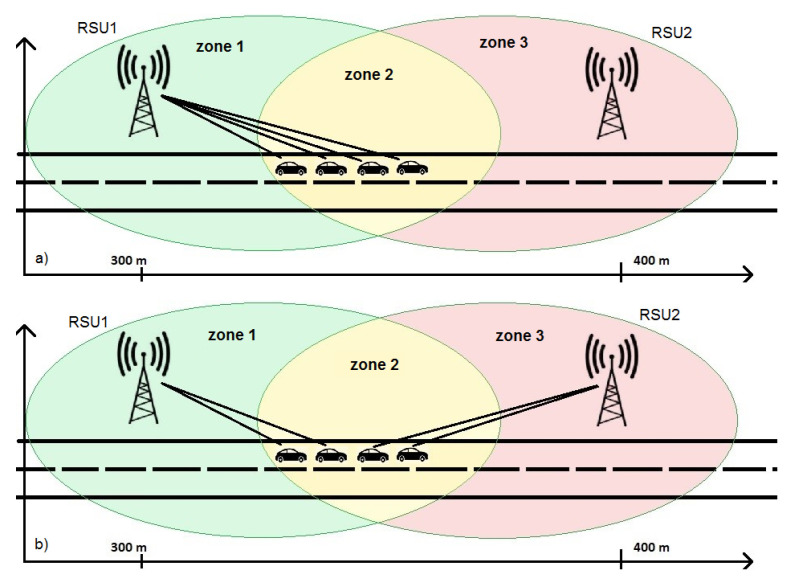
Switching mechanism: (**a**) without cell breathing-like approach; and (**b**) with cell breathing-like approach.

**Figure 6 sensors-21-03638-f006:**
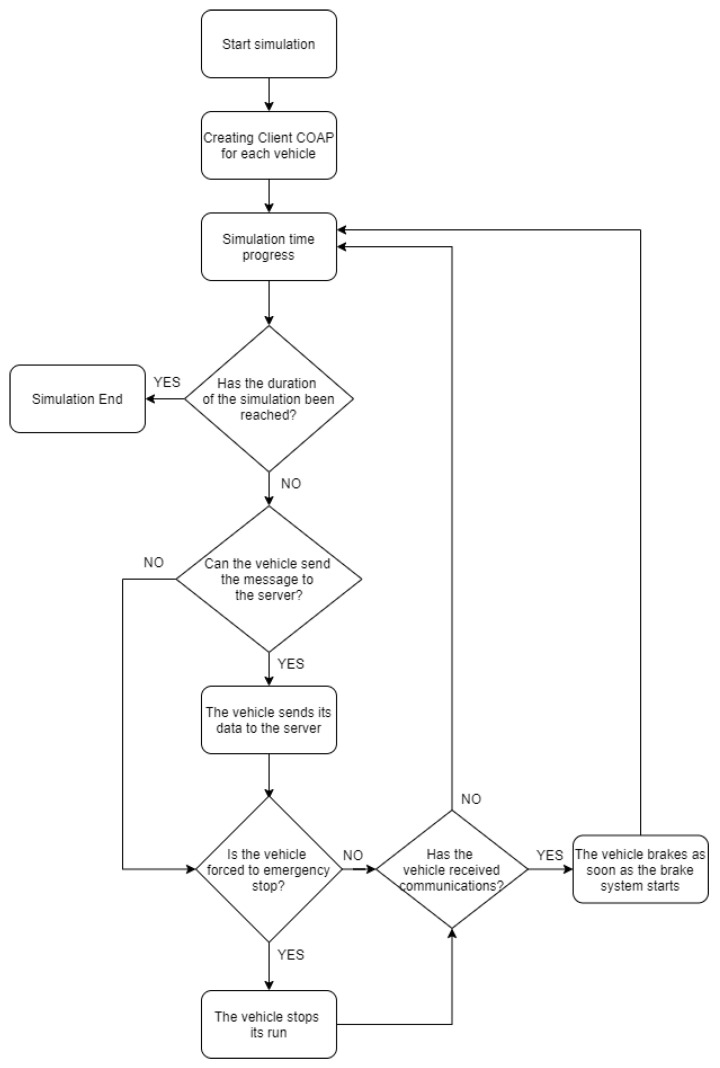
Flow chart of vehicle behavior.

**Figure 7 sensors-21-03638-f007:**
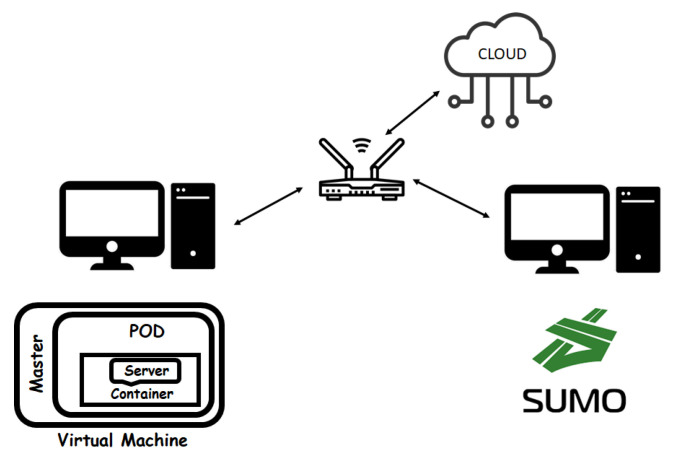
Considered scenario.

**Figure 8 sensors-21-03638-f008:**
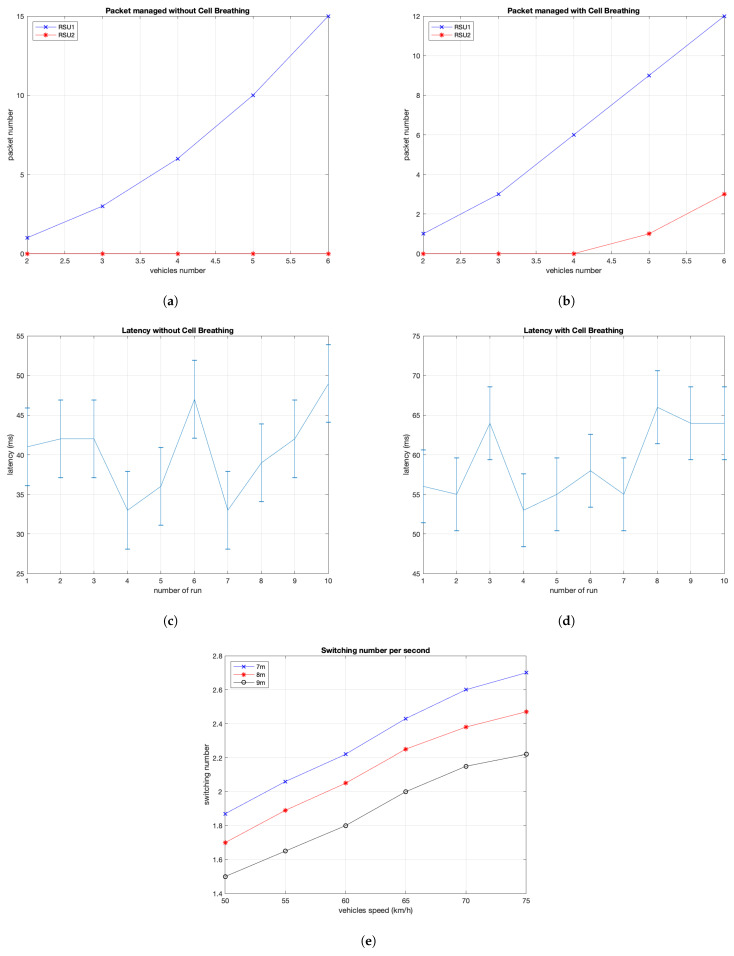
(**a**)Packets managed without cell breathing technique; (**b**) packets managed with cell breathing technique; (**c**) latency without cell breathing technique; (**d**) latency with cell breathing technique; and (**e**) switching per time unit.

**Figure 9 sensors-21-03638-f009:**
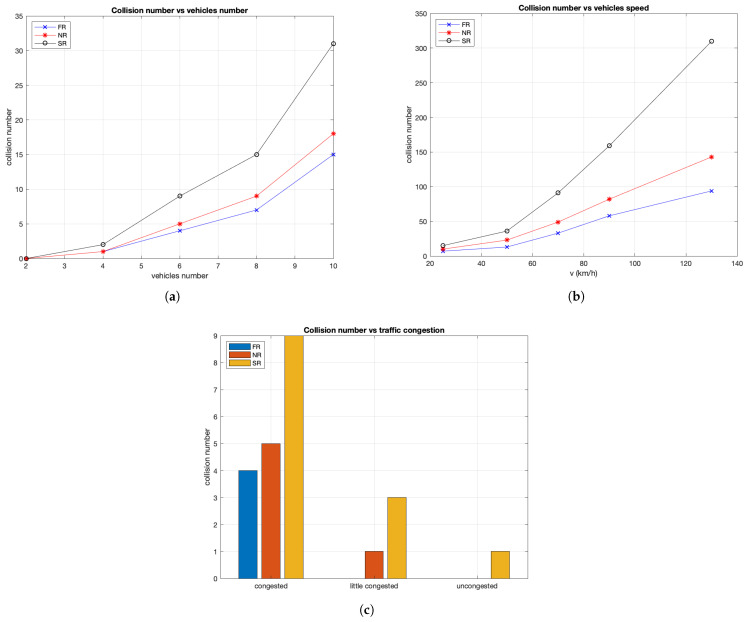
Number of collisions varying: (**a**) vehicle number; (**b**) vehicle speed; and (**c**) traffic scenario.

**Figure 10 sensors-21-03638-f010:**
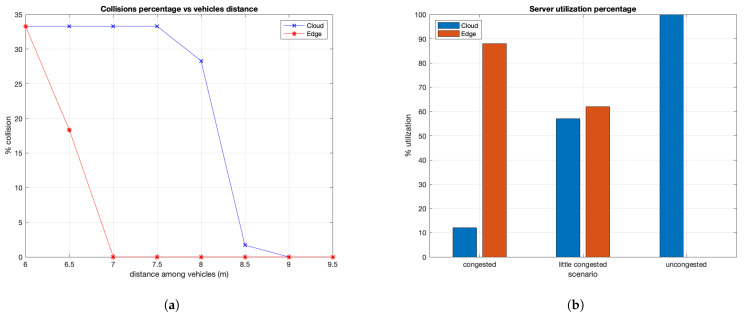
(**a**) Collision percentage vs. vehicles distance; (**b**) server utilization vs. traffic congestion.

**Figure 11 sensors-21-03638-f011:**
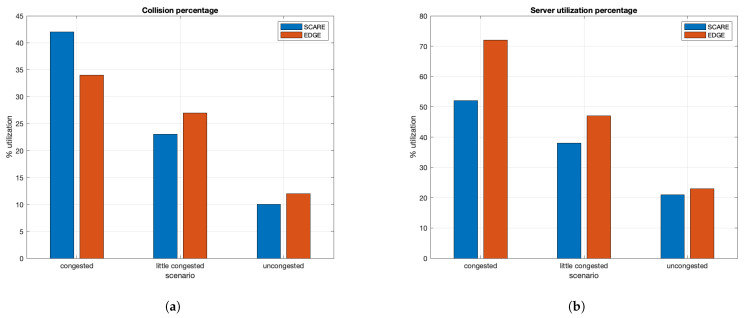
(**a**) Collision percentage vs. traffic congestion; and (**b**) server utilization percentage vs. traffic congestion.

**Table 1 sensors-21-03638-t001:** Comparison between MEC and Cloud computing in VANET.

	MEC	Cloud
location	near users	remote location
latency	low	high
mobility support	high	limited
decision making	local	remote
communication	real-time	no real-time
context awareness	yes	no
computing capability	medium/low	high

**Table 2 sensors-21-03638-t002:** Messages description.

Message Type	Message Description
*Type 0*	Periodic message that each vehicle sends to its reference server to update it on its speed and position
*Type 1*	Message exchanged among RSU devices and among RSUs and vehicles for notifying particular events
*Type 2*	Sent from the vehicle to the server to notify switching completion
*Type 3*	Message reply to a *Type 2* message to notify the vehicle dissociation
*Type 4*	Used to communicate the resizing of the coverage area of the server

**Table 3 sensors-21-03638-t003:** Speed and space of reaction.

*v* (km/h)	*v* (m/s)	$\Delta s_{r}$ (m)
25	7	7
50	14	14
70	19	19
90	25	25
130	36	36

**Table 4 sensors-21-03638-t004:** Speed and braking distance.

*v* (km/h)	$\Delta s_{f}$ (m)
25	4
50	16
70	31
90	52
130	110

**Table 5 sensors-21-03638-t005:** Symbols description.

Symbol	Description
*ID*	Vehicle identifier
*v*	Vehicle speed
*a*	Vehicle acceleration/deceleration
*d*	Distance among vehicles
$t_{lr}$	Latency plus reaction time

**Table 6 sensors-21-03638-t006:** Simulation parameters.

Parameter	Value
vehicles number	1–10
vehicle distance *d* (m)	6, 6.5, 7, 7.5,
	8, 8.5, 9, 9.5
vehicle speed *v* (km/h)	25–130
reaction time $t_{r}$ (ms)	750–1000–1500
latency time $t_{l}$ (ms)	1 (5G technology)

## Data Availability

Data sharing is not applicable to this article.

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
