# Peer review of "SCARE: A Novel Switching and Collision Avoidance pRocEss for Connected Vehicles Using Virtualization and Edge Computing Paradigm"

_sensors, 2021, doi:10.3390/s21113638_

Round 1
Reviewer 1 Report
- An abstract should be concise and cover the entire scope of the work in the mansucript. Rather than focusing on defining MEC and edge, it would be more appropriate if the outline of the work, the improvements it provides, as well as some quantified results are presented.
- A lot of significant portions of text show plagiarism from other online and published sources. These 2 seem to the main sources:
1) N. Nevigato, M. Tropea and F. De Rango, "Collision Avoidance Proposal in a MEC based VANET environment," 2020 IEEE/ACM 24th International Symposium on Distributed Simulation and Real Time Applications (DS-RT), Prague, Czech Republic, 2020, pp. 1-7, doi: 10.1109/DS-RT50469.2020.9213521.
2) Bitonti, L. and Tropea, M., 2020. Implementation and Simulation of Handover Techniques to Guarantee Service Continuity through Microservices at Edge.
- A lot of papers have been self-cited by the authors in this manuscript. Most of these works do not directly contribute to the development of this work. Please take corrective measures.
- Some typographical errors and grammatical changes need to be made in this paper. The authors should check carefully althrough this paper. For example, in page 2, lines 45-46 does not make any sense.
- The literature review is very shallow and needs to be significantly updated. More recent and relevant works need to be included here. Works on MEC have been pursued for quite some time now and therefore have a lot of content available which definitely need comparison and mention in this work. A paragraph on synthesis of information and identified gaps will be good to have at the end of section-2.
- The organization of the sections is confusing. I feel that sections 3.1 , 3.2....can be independent of reference scenario and thus not included under it. The authors need to massively overhaul the organization of this manuscript to enable readability and continuity of this work.
- are reaction times of 0.75-1.5 sec standard? Please provide references.
- a latency time of 1ms considering 5g comms is pursued in this work. Is there some standard reference for this work? Or, do the authors consider this for their simulation only? Either way, does this latency include noise effects, high-speed vehicles, effects of movement, etc.?
- The authors should compare their work against standard works, which are not their own, in this area. This will enable a reader to understand the effectiveness of the proposed approach and decide its merits. This would also allow the readers to understand the limitations of this approach.
- The conclusion sections needs overhauling. Also, please provide some quantified results and outline how it fares against already established approaches.
Author Response
We thank the reviewer for his comments that allow us to improve our manuscript. We will present reviewer questions and answer in the following
- An abstract should be concise and cover the entire scope of the work in the mansucript. Rather than focusing on defining MEC and edge, it would be more appropriate if the outline of the work, the improvements it provides, as well as some quantified results are presented.
We thank the reviewer for the suggestion. We have provided to change the abstract in order to have a more concise and focused abstract.
- A lot of significant portions of text show plagiarism from other online and published sources. These 2 seem to the main sources:
1) N. Nevigato, M. Tropea and F. De Rango, "Collision Avoidance Proposal in a MEC based VANET environment," 2020 IEEE/ACM 24th International Symposium on Distributed Simulation and Real Time Applications (DS-RT), Prague, Czech Republic, 2020, pp. 1-7, doi: 10.1109/DS-RT50469.2020.9213521.
2) Bitonti, L. and Tropea, M., 2020. Implementation and Simulation of Handover Techniques to Guarantee Service Continuity through Microservices at Edge.
We thank the reviewer for the observation. We have provided now to change some common parts differentiatin much more the manuscript contribution by our two previous published conferences.
- A lot of papers have been self-cited by the authors in this manuscript. Most of these works do not directly contribute to the development of this work. Please take corrective measures.
We thank the reviewer for the observation. We have provided to delete references not suitable for this work.
- Some typographical errors and grammatical changes need to be made in this paper. The authors should check carefully althrough this paper. For example, in page 2, lines 45-46 does not make any sense.
We thank the reviewer for the suggestion. In order to find and correct errors in the paper we have read it with attention.
- The literature review is very shallow and needs to be significantly updated. More recent and relevant works need to be included here. Works on MEC have been pursued for quite some time now and therefore have a lot of content available which definitely need comparison and mention in this work. A paragraph on synthesis of information and identified gaps will be good to have at the end of section-2.
We thank the reviewer for the observation. We have tried to include in the related work section new and more relevant papers. Moreover, as the reviewer suggests, we have introduced in section 2 a new paragraph, the paragraph 2.3 “Main paper contributions” in which highlighting point by point the main contributions provided by this work.
- The organization of the sections is confusing. I feel that sections 3.1 , 3.2....can be independent of reference scenario and thus not included under it. The authors need to massively overhaul the organization of this manuscript to enable readability and continuity of this work.
We thank the reviewer for the observation. We agree, we have re-organized section 3 and as consequence other sections:
- Section 3 has now only a sub section on the messages used in the considered VANET scenario, the old sub section 3.4
- A new section “Critical Parameters Design“ has been created from sub section 3.2 and 3.3
- Subsection 3.1 is now in section 7 “Software Implementation” as 7.1
- Now, the overall paper organization should appear more organic and more readable for readers.
- are reaction times of 0.75-1.5 sec standard? Please provide references.
We thank the reviewer for the observation. We have used typical reaction time values that are provided in literature. Thanks to the reviewer suggestion we have included in the references a couple of authoritative publications that used these values (the papers are included in the following). Moreover, we have also used another authoritative font that is a web portal of ASAPS that is a Police association (https://www.asaps.it/).
Droździel, P., Tarkowski, S., Rybicka, I., & Wrona, R. (2020). Drivers’ reaction time research in the conditions in the real traffic. Open Engineering, 10(1), 35-47.
Taleb, T., Benslimane, A., & Letaief, K. B. (2010). Toward an effective risk-conscious and collaborative vehicular collision avoidance system. IEEE Transactions on Vehicular Technology, 59(3), 1474-1486.
- a latency time of 1ms considering 5g comms is pursued in this work. Is there some standard reference for this work? Or, do the authors consider this for their simulation only? Either way, does this latency include noise effects, high-speed vehicles, effects of movement, etc.?
We are used this latency value as extreme case on the basis of some papers that report the value of 1ms for the possible latency time in a 5G communication. Clearly, the latency time depends on different factors and context, but in this work we wanted provide some indications in terms of number of collisions considering a very low latency provided by 5G technology. In the following a couple of papers that reports this possible value for latency in 5G.
Al-Saadeh, O., Wikstrom, G., Sachs, J., Thibault, I., & Lister, D. (2018, December). End-to-end latency and reliability performance of 5G in London. In 2018 IEEE Global Communications Conference (GLOBECOM) (pp. 1-6). IEEE.
Li, J., Chen, L., & Chen, J. (2021). Enabling technologies for low-latency service migration in 5G transport networks. Journal of Optical Communications and Networking, 13(2), A200-A210.
- The authors should compare their work against standard works, which are not their own, in this area. This will enable a reader to understand the effectiveness of the proposed approach and decide its merits. This would also allow the readers to understand the limitations of this approach.
We thank the reviewer for the suggestion. We have provided an analysis about the convenience of using, in a context of vehicular network in emergency context, a cloud or an edge server in order to show how the use of a hybrid mechanism is useful for guaranteeing an optimal collision avoidance mechanism and, at the same time, for avoid of overloading the edge server in RSU. To the best of our knowledge, and how it is possible to view in our extensive state of art, in the MEC and collision avoidance in VANET, nobody has yet analyzed a similar scenario showing how the human and the latency times can affect the collision number and percentage. So, for this work, our intention is to provide this useful indication. In the next and future works we could try to provide a comparison about different collision avoidance techniques proposed in literature, but at now, the paper aim is to provide indications on the best use of cloud/edge paradigm, showing, also, how an opportune switching mechanism can be used for better balancing vehicles inside RSU coverage. We hope that our explanation is sufficient for adequately responding to the reviewer observation.
- The conclusion sections needs overhauling. Also, please provide some quantified results and outline how it fares against already established approaches.
We thank the reviewer for the suggestion. We have provided to proofread the conclusion section in order to better explain the paper contribution highlighting the proposed mechanisms described and evaluated. The paper is a study and analysis of two important issues (load balancing and collision avoidance advertising) in vehicular networks and aims to give useful indication to the scientific community.
Reviewer 2 Report
This paper proposed a collision avoidance system based on MEC in a VANET environment to provide continuous service based on two modified switching mechanisms supported by micro-services at Edge. This research topic is timely needed for the development of intelligent transportation systems and the proposed ideas are interesting. In general, this paper is well-written. The following comments should be considered for further improvement.
1- The contributions are suggested to be highlighted point by point for readers to get a clear and better understanding of the present technology.
2- The work in the previous studies [22] and [23] should be briefly introduced. Then, the readers can better judge if the presented work is a reasonable extension of these two studies.
3- The reviewed literature on collision avoidance is insufficient. The related technologies in the following studies from the transportation community should also be reviewed and include: a real-time crash prediction fusion framework: an imbalance-aware strategy for collision avoidance systems; risk assessment based collision avoidance decision-making for autonomous vehicles in multi-scenarios.
4- Typo: breaking should be braking. Please carefully proofread the manuscript before resubmission to avoid typos and syntax errors.
5- The font size in figure 5 is too small for a clear presentation.
Author Response
We thank the reviewer for his comments and suggestions. We presented below questions and answers to reviewer observations.
This paper proposed a collision avoidance system based on MEC in a VANET environment to provide continuous service based on two modified switching mechanisms supported by micro-services at Edge. This research topic is timely needed for the development of intelligent transportation systems and the proposed ideas are interesting. In general, this paper is well-written. The following comments should be considered for further improvement.
1- The contributions are suggested to be highlighted point by point for readers to get a clear and better understanding of the present technology.
We thank the reviewer for the suggestion. We have added a paragraph 2.3 titled “Main paper contributions” in order to highlight point by point the contributions of this paper.
2- The work in the previous studies [22] and [23] should be briefly introduced. Then, the readers can better judge if the presented work is a reasonable extension of these two studies.
We thank the reviewer for this suggestion. We have introduced a brief description of the two previous paper that are at the basis of this work. We have added this description in the paragraph 2.3 where also we have highlighted the paper contributions.
3- The reviewed literature on collision avoidance is insufficient. The related technologies in the following studies from the transportation community should also be reviewed and include: a real-time crash prediction fusion framework: an imbalance-aware strategy for collision avoidance systems; risk assessment based collision avoidance decision-making for autonomous vehicles in multi-scenarios.
We thank the reviewer for the observation. We have provided to extend the reviewed literature adding also the suggested and interesting works.
4- Typo: breaking should be braking. Please carefully proofread the manuscript before resubmission to avoid typos and syntax errors.
We thank the reviewer, we have provided to read more carefully the overall work in order to correct as many as possible errors.
5- The font size in figure 5 is too small for a clear presentation.
We have provided to change the figure in order to make it clear and readable.
Reviewer 3 Report
Dear Authors,
Very nice paper, well written, also present the algorithms developed. In addition the topic is a trend in intelligent transportation system.
Just check the English, and avoid using first person pronouns.
Greetings :)
Author Response
We thank a lot the reviewer for his appreciation of the work.
Round 2
Reviewer 1 Report
I would have been fully satisfied if the authors could have included some form of comparative analysis, fully or in parts. However, I agree with the authors' responses to my previous queries and feel like I do not have anything to add further.